# Main Agro-Ecological Structure: An Index for Evaluating Agro-Biodiversity in Agro-Ecosystems

**Ingrid Quintero [1,2,\*], Yesica Xiomara Daza-Cruz [2] and Tomás León-Sicard [2]**

[1] Programa Doctoral en Agroecología, Facultad de Ciencias Agrarias, Universidad Nacional de Colombia, Carrera 30 # 45–03 Edificio 500, Bogota 111321, Colombia

[2] Instituto de Estudios Ambientales, Grupo de Estudios Ambientales Agrarios, Universidad Nacional de Colombia (IDEA/UN), Calle 44 # 45–67, Unidad Camilo Torres Bloque B2, Bogota 111321, Colombia

\* Correspondence: iquinteros@unal.edu.co; Tel.: +57-(1)-316-5000 (ext. 19063)

**Abstract:** The agro-biodiversity present in agro-ecosystems is fundamental in guaranteeing sustainability and resilience. However, there are very few approaches for evaluating it and, even less, ones that include indicators to analyze the influence of the structural and spatial configuration of a landscape in order to favor agro-biodiversity connectivity to productive systems. There are also no proposals that include management and conservation practices, and the producer (farmer)'s perceptions, awareness and ability to favor it on their farm. The Main Agro-ecological Structure (MAS) is a new proposed index to describe the agro-biodiversity of agro-ecosystems, including these topics and comprising 10 criteria and 29 indicators. Connection with the main ecological structure of the landscape (CMESL), extension of external connectors (EEC), diversity of external connectors (DEC), extension of internal connectors (EIC), diversity of internal connectors (DIC), land use (LU), management practices (PM), conservation practices (CP), perception, awareness and knowledge (PAK) and action capacity (AC) are the criteria that make up the index. Methodologies for its evaluation are also described, and a possibility of adapting certain indicators according to the ecological and cultural contexts where the farms are located is discussed. The possibility of relating agro-biodiversity, evaluated using the index, to other agro-system attributes or properties would allow for one to consider its importance in the multidimensional sustainability of farms, thought to be the greatest advantage for its application.

**Keywords:** agroecology; Andean agroecosystems; agro-biodiversity index; cultural dimensions; farmer awareness; farmer perceptions

## 1. Introduction

Conventional agricultural and livestock handling practices based on species monoculture and improved races, the intensive use of agrochemical substances, and the intensive mechanization implemented in agro-systems all threaten agro-biodiversity [1,2]. Not only are the species, races and varieties cultivated directly, but the diversity essential for ecological processes that intervene in and benefit production: nutrient recycling, fertilization, and biological control, among others [3–6], are as well.

Despite the importance of agro-biodiversity and its multi-scale spatial dependence (local, regional, national, even global) for the development of agriculture, very little has been done to evaluate it. Recently, International Biodiversity proposed the Agro-biodiversity Index (ABD) for evaluating a country's agro-biodiversity [7]. Indicators to evaluate the diversity or food groups, species and varieties, (Pillar I: the functional diversity (pollinizing) of soil, cultivated plants and breeding animals, and (Pillar II) the genetic diversity of plants and animals with the support of the ethno-botanical knowledge of the communities (Pilar III), are proposed [7,8].



On a farm scale, certain steps have been taken to evaluate agro-biodiversity. Leyva and Lores (2012, 2018) propose four indicators to represent species wealth, classified as different types or indexes that relate to species biodiversity for human, animal, and microorganism food and other aspects, not related to nutrition, that have become the 14 indicators of the Agro-ecosystem Diversity Index (ADI) [9,10].

Vazquez et al. (2014) classified the different levels of biodiversity in an agro-ecosystem from a functional perspective into associated, introduced and auxiliary production using indicators that evaluate the elements, designs and management of the productive system and, thus, determine the Biodiversity Management Coefficient (BMC). In this manner, the complexity of interactions among these components is evaluated to produce an indicator called Farm Biodiversity Density Interactions (FBDI) that expresses the possible number of interactions [11].

Analysis of these approaches reveals that the magnitude of agro-biodiversity elements and how they interact are the most valued aspects. However, this does not include those elements related to the spatial structure of this diversity within the farm and the landscape, or how they are affected by the cultural characteristics of the owners. These two aspects become more important since the functional processes within agro-ecosystems to produce benefits for producer and society depend on the relationships between the landscape matrix (agro-landscape) and the farm, characteristics (agro-biodiversity structuring and planning) on which owner or producer decisions depend.

In answer to the lack of information from the analysis, the Main Agro-ecological Structure (MAS), formulated by Leon-Sicard (2010; 2021) and Leon-Sicard et al. (2018), is defined as an environmental descriptor for farms. This index comes about as an attempt to characterize farm or agroecosystem agrobiodiversity from a complex environmental approach. This would include ecosystemic aspects, related to the spatial and structural configuration of vegetal connectors on farm perimeter and the surrounding agro-landscape, as well as cultural aspects that include the agro-biodiverse character of the administrative practices of the productive system, perception, conscience–knowledge, and the producer's diverse capacities for action in order to promote agrobiodiversity on the farm [12–14].

Based on Leon-Sicard (2014) and Leon-Sicard et al.'s (2018) research, the authors restructured criteria, indicators and methodology indexes in order to produce a proposal where MAS is an agro-ecosystem biodiversity descriptor. Twenty-nine (29) indicators were structured from the 10 final criteria, as is presented throughout this document, partially consigned in Leon-Sicard (2021) [12,13,15].

## 2. Indicators Selection

In October 2018 and May 2021, the initial MAS 10 criteria and 13 indicators were evaluated and restructured based on group discussions of the Environmental Agrarian Studies Group, National University of Colombia, considering different agro-biodiversity dimensions, such as ecological and spatial attributes of biological connectors, diverse management practices in the phases of agricultural and livestock production, conservation practices, farmer skills and perceptions to increase their agrobiodiversity. After preselection of indicators and value scales, they were submitted to consideration by specialists in related disciplines (agroecology, psychology, sociology, ecology) and refined with a documentary review approach. Ten criteria and twenty-nine final indicators were selected to conform a new MAS proposal.

MAS criteria have a generalized structure to permit comparisons among farms with a high variety of management styles, agro-ecosystem matrixes, and cultural contexts where agro-biodiversity is structured. However, the index is still evolving so that some specificities that reflect local conditions can still be included. The indicators are additive in character and would be included in each of the 10 established criteria. This would also easily permit the incorporation of other indicators without the index losing its descriptive and comparative nature [12,15].

### 3. Development of MAS as Agro-Biodiversity Index

*3.1. Connection with the Main Ecological Structure of the Landscape (CMESL)*

This parameter describes the spatial relationships of the agro-ecosystem with the natural elements of the surrounding landscape or main ecological structure (According to van der Hammen and Andrade (2003, pg. 17), main ecological landscape structure is "the set of natural and semi-natural ecosystems that have a localization, extension, connections and state of health that guarantee the integrity of the biodiversity of the location, providing ecosystem services as a measure to guarantee the satisfaction of the inhabitants basic needs and perpetuation of life", especially with vegetation fragments (forests, bush vegetation, secondary vegetation, gallery forests or other types of biological vegetation connectivity) and bodies of water, whether aquatic ecosystems or artificial reservoirs. The reason behind this approach is that the functional processes of agro-biodiversity within crops are affected by landscape composition and configuration (types and quality, quantity, proportion and distance among habitats) [16,17].

In agricultural mosaics or complexes (heterogeneous landscapes in terms of composition, structure and spatial arrangement of vegetation elements) where there is a large proportion of natural or semi-natural habitats that act as a refuge for certain organisms, ecological dynamics can proceed from natural areas to agro-ecosystem matrix, or from those agro-ecosystems toward the forests or natural vegetation, responding to the connectivity and permeability of its matrix, as well as to the spatial disposition of water [18–20]. In a high-quality matrix, bidirectional transit increases the possibility of functionally important mobile species persisting in the landscape and of key functions of their production and productivity being maintained in the agro-ecosystems [18,21,22].

In these complex landscapes, not only ecological functions seem to be favored economic benefits can also increase since productivity of crops such as corn, soy, and winter wheat may be 20% higher than in simplified landscapes [23].

Establishment of Area of Influence in High Quality Matrixes (AI)

In order to determine the relationships between the agro-ecosystems and the previously considered landscape elements, a zone of influence must be established for the farm. Wiegand et al. (1999) considered that an appropriate landscape metric should not only characterize the different types of habitats but must also consider the spatial relationships among and within habitat types as a function of distance. For this, they developed a ring statistic with a radius (r) to characterize spatial landscape structure in function of the "organism perception" (called perceptual distance) of each of the habitats present there, beginning with the site where the said organism was found [24].

It is difficult to propose a single value for adapting this concept to the effect or influence of the landscape on agro-ecosystem components in terms of productivity, adaptability, or resilience, among other properties affected by agro-biodiversity. To simplify this, Leon-Sicard et al. (2018) and Leon-Sicard (2021) proposed defining a circle with a radius measuring the double of the longest side of the farm [12,15]. In all cases, this would permit having an extension of land proportional to the area of each farm and emphasize the importance and centrality of the agro-ecosystem in relation to the landscape. At the same time, it would permit defining a zone or area of influence over the farm and vice versa (Figure 1, Equations (1) and (2)).

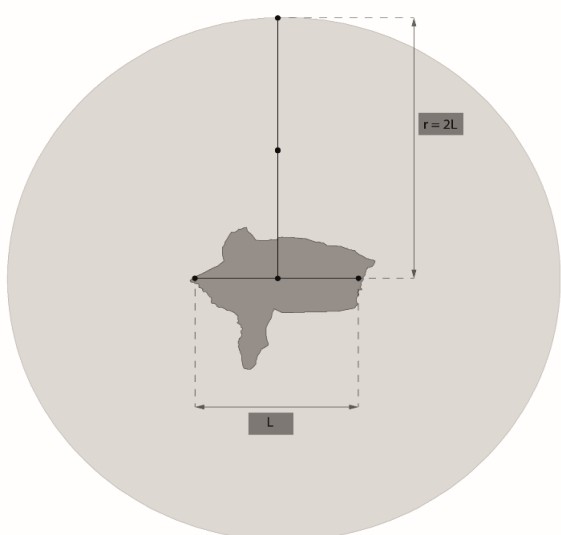

**Figure 1.** Scheme that reflects selection of farm area of influence (Source: Quintero et al., 2022).

$$r = 2L \tag{1}$$

$$AI = \pi r^2 - AF \tag{2}$$

where $L$ = length of farm's longest side, $r$ = radius of area of influence, measured from center of farm, $AI$ = area of influence, $AF$ = total farm area.

In the CMELS, Leon-Sicard (2021) proposed including three basic indicators in order to evaluate the average distance between vegetation fragments ($DFr$) and bodies of water ($DBw$), taking the point closest from this coverage to the center of the farm located in its area of influence ($AI$) into account [12]. This will permit considering the relative distances of these elements within the agro-landscape defined by $AI$ as well as the total area occupied by vegetation fragments and bodies of water ($AFrBw$) in the $AI$. These relationships are presented in Equations (3)–(5).

$$DFr = \left( 1 - \left( \frac{\left( \frac{\sum_1^m dFr_t}{m} \right)}{r} \right) \right) \times 10 \tag{3}$$

where $DFr$ = indicator of the average distance of fragments ($Fr$) in the $AI$ at the center of the farm, $dFrt$ = distance of fragment type, $m$ = number of vegetation fragments, $r$ = radius that defines the farm's area of influence.

$$DBw = \left( 1 - \left( \frac{\left( \frac{\sum_1^n dBw_j}{n} \right)}{r} \right) \right) \times 10 \tag{4}$$

where $DBw$ = indicator of the average distance of the bodies of water ($Bw$) at the $AI$ at the center of farm, $dBwj$ = distance of body of water type j, $r$ = radius of farm area of influence, $n$ = number of bodies of water.

$$AFrBw = \frac{\sum_1^m AFrt_i + \sum_1^n ABw_j}{AI} \times 10 \tag{5}$$

where $AFrBw$ = indicator of total density of fragments and body of water in the $AI$ of the farm, $AFrt$ = area of fragments type t, $ABw_j$ = area of bodies of water type j, $AI$ = farm area of influence.

$$CMELS = \frac{DFr + DBa + AFrBa}{3} \qquad (6)$$

The values obtained for these criteria and for the following nine criteria can be interpreted according Leon-Sicard (2021) [12].

### 3.2. Extension of External Connectors (EEC)

These criteria refer to the presence and length of vegetation connectors (live fences, fences, hedges, linear or riparian forests associated with bodies of water) found on the farm perimeter. They are evaluated as a percentage of the linear extension of these vegetation connectors along the total farm perimeter according to Equation (10).

$$EEC = \frac{\sum_1^n (LCV_i)}{PF} \times 10 \qquad (7)$$

where $EEC$ = indicator of external extension connectors, $LCV_i$ = length of each connector $i$ with vegetation $PF$ = perimeter of farm.

Again, these criterion considers landscape spatial connectivity toward the farm and vice versa, finding a spatial element in the perimeter that, in functional terms, can increase or diminish it. However, the patches of vegetation within a heterogeneous matrix can be considered as biodiversity connectors to and from the productive system. Thus, it would be an error not to consider them when measuring farm connectivity. In conclusion, patches maintained along the perimeter and inside the farms are important agro-biodiversity connectors [16].

### 3.3. Diversity of External Connectors (DEC)

This criterion describes the quality of these previously identified vegetation connectors based on an analysis of their richness (Table 1) and vertical structure (Table 2), fundamental properties of vegetation, and diversity [25,26]. For average species richness, ranges of diversity assessment that also apply to internal connectors were adjusted according to the average plant species richness that can be contained in a Colombian sub-Andean forest with more than 40 species per locality, according to Cuatrecasas (1958) [27]. The classification of these vegetative strata proposed by Rangel and Lozano (1986) and adjusted for Andean ecosystems that can be found in rural landscapes considers the following strata: herbaceous 0.3–1.5 m.; bushy 1.5–5 m.; sub-arboreal or small trees 5–12 m.; upper arboreal >20 m.; and emerging 20–35 m.

**Table 1.** Weighing of levels of internal and external connector richness in the agro-ecosystems (Source: modified from [12]).

| | Classification Rr | Value |
|---|---|---|
| $R_{ma}$ | 31 or more species. | 10 |
| $R_a$ | Between 21 and 30 species. | 8 |
| $R_m$ | Between 11 and 20 species. | 6 |
| $R_b$ | Between 5 and 10 species. | 3 |
| $R_{mb}$ | With less than five species. | 1 |

**Table 2.** Weighing of internal and external connector vegetative strata in the agro-ecosystems (Source: modified from Leon-Sicard 2021).

| | Classification $E_e$ | Value |
|---|---|---|
| $E_{ma}$ | More than five vegetative strata. | 10 |
| $E_a$ | Four vegetative strata. | 8 |
| $E_m$ | Three vegetative strata. | 6 |

| $E_b$ | Two vegetative strata. | 3 |
| $E_{mb}$ | Only one vegetative strata. | 1 |

Given the methodological complexity and difficulty in evaluating the effects of these two attributes independently, it is thought that through greater diversity, the connectors could permit greater movement and a permanent or potential habitat in the agricultural landscape for certain species [28], which would fulfill an important function in productive systems. This aspect has been widely documented in the literature and is exemplified in studies by Garibaldi et al. (2011) in their observations of pollinating bee foraging, nesting and recruitment sites in hedges, reed corridors, and flowering plants along cultivated plots, and in studies by Letorneau et al. (2011) of the increased biological control of pests when species present along the edges of diversified crops [29,30].

Most ecological landscape research does not specify the differences between connectors in relation to their location on the farms, in other words, whether they are part of farm limits or whether they separated productive systems or infrastructure (houses, corrals, deposits, sheds). This separation tends more toward owner interest, or the importance given to their productive spaces or those of their neighbors when considering aspects of biodiversity or not.

The description of external connector wealth and stratification is developed from Equations (8) and (9), and the criteria are calculated according to Equation (10).

$$RiEC = \frac{\sum_1^n (LVC_r * R_r)}{LCV_{total}} \tag{8}$$

where *RiEC* = indicator of exterior connector species richness, $LCV_r$ = length of vegetation connector with type r richness classification, $R_r$ = weighing of type r richness (Table 1), $n$ = number of richness types found along the perimeter, $LCV_{total}$ = total length of perimeter connectors.

$$EsEC = \frac{\sum_1^n (LCV_e * E_e)}{LCV_{total}} \tag{9}$$

where *EsEC* = indicator of exterior connector vertical stratification, $LCV_e$ = length of vegetation connector with type e stratification classification, $E_e$ = weighing of type e stratification (Table 1), $n$ = number of stratification types found along the perimeter, $LCV_{total}$ = total length of perimeter connectors.

$$DEC = \frac{RiEC + EsEC}{2} \tag{10}$$

### 3.4. Extension of Internal Connectors (EIC)

This criterion evaluates the length and presence of the interior connectors found on the farm, separating the different types of productive systems, whether agricultural crops or pastures. They are evaluated as percentage of the linear extent of vegetation connectors over the total length of the interior farm divisions that separate productive areas based on Equation (11).

Vegetation connectors on crop peripheries would facilitate functional agro-biodiversity movement toward productive systems [31], aspects that also depend on connector quality (see Criteria III and V) and on organisms related to this function. Additionally, they can supply other producer benefits, such as mitigating nitrogen and phosphorous loss by runoff, reduce water channel contamination by pesticides and other agro-chemicals, act as a buffer against extreme microclimatic conditions, avoid erosion and offer wood, food and forage for cattle [32–36]. Nevertheless, interior connectors on most Colombian farms are composed of barbed-wire fences that offer no eco-systemic or cultural benefit besides serving as a physical limit to the movement of different organisms.

$$EIC = \frac{\sum_1^m (LI_j VC)}{LID} \times 10 \tag{11}$$

where $EIC$ = indicator of interior connection extent, $LI_j VC$ = length of each interior division that is a vegetation connector, $LID$ = total length of interior divisions.

### 3.5. Diversity of Internal Connectors (DIC)

This criterion considers species richness and vertical vegetation structure associated with the connectors within the farm and that separate crops and pastures. The indicators were re-evaluated using Equations (12) and (13) and can be carried out using Tables 2 and 3. Total of criterion value is developed in Equation (14).

$$RiIC = \frac{\sum_1^n (LCVI_r * R_r)}{LCVI_{total}} \tag{12}$$

where $RiIC$ = indicator of interior connector species richness, $LVCI_r$ = length of vegetation connector with type r richness classification, $R_r$ = pondering of richness type $r$ (Table 2), $n$ = number of types of richness found in interior connectors, $LCVI_{total}$ = total length of internal connectors.

$$EsCI = \frac{\sum_1^n (LVCI_e * E_e)}{LCVI_{total}} \tag{13}$$

where $ICEs$ = indicator internal connector vertical stratification, $LVCi$ = length of vegetation cover with type $i$ stratification classification, $E_i$ = weighing of stratification type i (Table 2), $n$ = number of types of stratification found in interior connectors, $LCVI_{total}$ = total length of internal connectors.

$$DCI = \frac{RiCI + ICEs}{2} \tag{14}$$

In order to evaluate Criteria 1, 2 and 4, Leon-Sicard (2021) proposed the use of high-resolution satellite images where farms of interest are projected, whether they be free use for academic purposes or, in their defect, for commercial purposes [12]. When there are none, one solution is to use of drones and photogram techniques. The methodology developed in Quintero et al. (2022) details the procedures for these ends [37]. With this basic information, it is possible to obtain landscape metrics using several GIS tools, national cadastral systems, spatial platforms such as Google Earth, or software such as QGis or ArcGIS. In case this, equipment is not available, qualitative methods of participative cartography and observation can be used.

In order to evaluate Criteria 3 and 5, the floristic and phyto-sociological characterization of the vegetation using different sized plots and transects according to connector characteristics and complexity requires a botanist. In case a professional botanist is not available, taking advantage of the experience of farmers and owners to recognize species, uses and other knowledge associated to this diversity to enrich and complement the study is recommended [38]. The quality of farmer observation is a powerful tool for separating morphotypes and morphospecies as taxonomically recognizable units [39] and to organize standardized databases that permit interpretative and comparative analysis.

If the agro-ecosystems contain large areas with connectors and some areas of the perimeters are not very accessible for floristic evaluation, a process of photo-interpretation of coverage based on high-resolution spatial images and the support of previously characterized field control points would contribute to the analysis of the entire farm.

### 3.6. Land Use (LU)

Soil uses are human designations for different portions of land with an interest function and represent cultural forms of territorial appropriation originated by the intentions,

history and knowledge of the person (whether it be owner or possessor of the space possessed) who makes the decision regarding its function which, therefore, modifies the agro-biodiversity [40]. This transformation affects functional ecosystem processes (and biodiversity), which oblige species to adapt or, in some cases, lead to their local extinction, depending on the intensity of the disturbance and the life histories of those species [41–43]. When taking soil characteristics into account, this criterion grants special representation to those who favor agro-biodiversity on the farm in the form of multi-crops, agro-forestry and silvo-pasture systems, or to even the same natural ecosystems that can be projected for agro-tourism and conservation, as expressed by Equation (15).

$$LU = \frac{\sum_1^n A_{ABj}}{AF} \times 10 \tag{15}$$

where $LU =$ soil use indicator, $A_{ABj} =$ area of type j uses that benefit agro-biodiversity $AF =$ total farm area.

Participative cartography and the support of the remote perception images obtained (satellite images, aerial photographs or ortho-mosaics from drone flights) allow owners to contribute to delimiting the areas over their physical versions. With the support of control points, geo-positioned by the researchers on several sectors of the farm, it is possible to use a triangulation methodology [44] and classification according to CORINE (Coordination of Information System on Environment) land cover methodology categories.

*3.7. Management Practices (PM)*

3.7.1. Agriculture Management Practices (aMP)

To eliminate chemical inputs and increase energy efficiency, it is necessary to restore biodiversity in the agricultural landscape and on the farm [45]. During the various phases of production, seed selection, soil preparation, fertilization, and phytosanitary management, it is possible to implement practices that favor and enhance agro-biodiversity.

Through the use of native seeds, genetic diversity is recognized and promoted; alimentary sovereignty is assured by adapted and resilient agricultural products; and bio-cultural patrimony in association with cultural practices that include seed selection, adaptation, conservation and interchange are protected [46,47].

Those seeds or races that have been adapted by native communities or small farmers under extreme climatic regions (varieties of corn that resist long periods of draught in Atacama or Kenya desert areas, for example) are important since they represent a viable alternative for regions that may experiment strong fluctuations in climate and, thus, contribute to the alimentary security and sovereignty of peoples in times of climatic change, environmental stress, natural disasters, and abundance of pests or illnesses [48,49].

Conservation agriculture and agro-ecology, as well, understand soil as the essential living element for the development of strong, healthy plants, and thus, its practices are directed toward creating conditions to minimize disturbances and protect and feed edaphic biodiversity (see aMP criterion). In a combination of simultaneous effects, they act positively on the stability of the edaphic biota, affecting decomposition processes, nutrient cycling, bioturbation, soil aggregate stability, apparent density, among others, to maintain their agricultural vocation [50–53].

Vegetal diversification, through the restoration of natural control among species where the integration and handling of weed plants is promoted, also contributes to the regulation of pests through the restoration of natural control among species, an aspect that is not appreciated by the conventional agricultural approach or use of agrochemicals [54,55].

Finally, after completing all the agricultural management processes needed to harvest the food (including medicinal and aromatic plants), the producer obtains a variety of products that make up part of their diet and contribute to their alimentary and nutritional

security by supplying basic foods as well as contributing to the conservation of agro-biodiversity in situ, not only of species but also of traditional crop varieties (landraces) [56].

On small farms located in the Andean Region of Colombia where family agriculture is developed, it is possible to find from 25 to 141 species (80 on average and more than two varieties in many cases) of plants for auto-consumption. Here, crop diversification favors conservation process production and can produce economic benefits since the excess can be sold to buy other foods that the farm does not produce. This offers opportunities for innovation and gives aggregate value to certain products through the processing and commercialization of little used foods with high nutritional, decorative, or domestic value and favors the conservation of the cultivated plant gene bank [57,58].

In Table 3, evaluation categories, based on these considerations and the five indicators proposed by Leon-Sicard (2021), are established for agricultural management practices: seeds (SEe), soil preparation (SoP), fertilizing (FEr), phytosanitary management (PyM) and crop diversification (CrD).

**Table 3.** Descriptors, evaluation and values of the criteria indicators for Agricultural Management Practice (aMP) (Source: modified from [12]).

| Indicator | Description | Evaluation Categories | Value |
|---|---|---|---|
| Seeds (SEe) | Type, production and conservation. | Own seed, ecological/ancestral, diverse and produced locally. Conserved through ecological practices. | 10 |
| | | Acquired seed, ecological/ancestral, diverse and obtained locally. Conserved through ecological practices. | 8 |
| | | Acquired seed, organic, diverse, and not obtained locally. Conserved through chemical procedures. | 6 |
| | | Conventional seed, not diverse (hybrids) and not obtained locally. Conserved by chemical procedures. | 3 |
| | | Transgenic seed. | 0 |
| Soil preparation (SoP) | Type of tillage, intensity, Use of conservation agriculture practices. | Zero plowing. Low intensity labor. Agricultural conservation practices: green fertilizer, coverage or mulch, harvest residue management, stubble and/or fallow. | 10 |
| | | Reduced tillage. Non intensive labor. With or without soil conservation practices. | 8 |
| | | Reduced tillage (chisel). Medium intense labor. Without soil conservation practices. | 6 |
| | | Conventional tillage (plows, rakes, dredges). Intensive labor. A soil conservation practice. | 3 |
| | | Conventional tillage. High intensity labor. Without soil conservation practices. | 0 |

| | | | |
|---|---|---|---|
| Fertilization (FEr) | Types of manure and fertilization, rotation, Complementary practices. | Organic fertilizers produced on farm: compost, manure, humus, green fertilizer, bio-fertilizers, microbe preparation, worm compound. High rotation. With complementary practices (use of mulch, fallow). | 10 |
| | | Purchased organic compounds. High rotation. With complementary practices. | 8 |
| | | Organic fertilizers mixed with chemical fertilizers. High to medium rotation. Few complementary practices. | 6 |
| | | Chemical fertilizers with low dosage. Little rotation. Some complementary practices. | 3 |
| | | High doses of chemical fertilizers. Without rotation. With no complementary practices. | 0 |
| Phytosanitary management (PyM) | Weeds management. Complementary practices. Biological, mechanical or chemical pest control. | Ecological handling of weeds. Use of complementary practices: bioles, slurry, hydrolates, push–pull systems, accompanying crops. Mechanical and biological controls are used. Pesticides are not used. | 10 |
| | | Ecological handling of weeds. Few complementary practices. Biological and mechanical controls. Pesticides are not used. | 8 |
| | | Ecological handling of weeds without complementary controls. Mechanical and biological controls. Application of recommended pesticides in low doses. | 6 |
| | | Manual weed eradication, some complementary practices, mechanical controls. Application of pesticides in recommended doses. | 3 |
| | | Chemical eradication of weeds. Elimination of habitats without complementary processes. Mechanical or biological controls. Application of pesticides in higher doses than the recommended. | 0 |
| Crop diversification (CrD) | Species and variety cultivated for human consumption. | More than 60 species, where at least two or more varieties of three species are cultivated (native and commercial). | 10 |
| | | 60 or more species where at least two or more varieties of two species are cultivated (native and commercial). | 8 |

| | Between 30 and 60 species, with no native varieties. | 6 |
| | Between 5 and 29 species, with no native varieties. | 3 |
| | Less than 5 species, with no native varieties. | 0 |

Once the previously presented indicators are described, a final qualification is defined for agricultural management practices according to Equation (16).

$$aMP = \frac{SEe + SoP + FEr + PyM + CrD}{5} \tag{16}$$

3.7.2. Livestock Management Practices (lMP)

This criterion evaluates the processes relevant to the livestock system, where soil and planting bed preparation (SoP), forage system arrangement (ASi), pasture rotation (PRo) and sanitary management (SaM), and that of animal drinking water to guarantee their health and nutrition are included. During the developmental phases of sustainable or ecological livestock systems, besides maintaining the economic viability of the herd to potentiate agro-biodiversity functions in the productive system, their practices can be directed (Table 4).

**Table 4.** Descriptors, evaluation and values of the cattle raising management practice (lMP) (Source: [12]).

| Indicator | Description | Evaluation Categories | Value |
|---|---|---|---|
| Soil preparation (SoP) | Type and intensity of tillage, manure, fertilizers or corrections, complementary practices | Zero tillage or very low intensity labor: direct planting. Use of corrections and organic matter. With complementary practices: forage associations with previous (potato, pea, corn, and/or bean) or accompanying crops, application of mycorrhiza, conservation of big trees and palms in the paddocks. | 10 |
| | | Minimum tillage or low intensity labor: planting in grooves, or vertical (use of light mechanization with furrows), mechanical or animal traction sowers. Use of corrections and organic fertilizer. With or without complementary practices. | 8 |
| | | Conventional tillage or medium intensity labor: sowing breaking up, loosening and chopping the ground (use of light mechanization or manually with a hoe). Low mineral or chemical fertilization, in lower doses than recommended. With or without complementary practices. | 6 |
| | | Conventional tillage or high intensity labor: mixed sowing or manually with hoe. Chopping and rechopping the soil (use of heavy machinery or manually with hoe). Medium or sporadic chemical fertilization according to recommendations. Without complementary practices. | 3 |
| | | Conventional tillage or very high intensity labor: sowing by deeply digging and turning over the soil (use of heavy | 0 |

| | | machinery). Very high frequent or chemical fertilization, higher than recommended doses. With no complementary practices. | |
|---|---|---|---|
| System arrangement (SiA) | Silvo-pasture system, diversity of grasses and legumes, dispersed trees, forage banks (Although live fences and windbreak curtains are mentioned in traditional evaluation of agroforestry type, they are not included in this table because their evaluation was already carried out in terms of internal and external connectors) | Intensive silvo-pasture system (iSPS) with several additional silvo-pasture systems in more than 75% of the farm´s productive area. High diversity of forage grasses (tussocks or stolonifers) and creeping legumes. Exist mixed forage banks. | 10 |
| | | iSPS and/or two additional silvo-pasture systems on less than 75% of the farm's productive area. High diversity of forage grasses (two or more of tussock growth such as stolonifers). Trees and bushes (for different uses, including forage) high density dispersion ($\geqslant$25 individuals ha$^{-1}$). Two (2) additional forage matter exist. | 8 |
| | | Without iSPS or other silvo-pasture systems. Medium forage grass diversity. Combination of two forage grasses where growth type does not matter, low tree and bush density (<25 individuals ha$^{-1}$) but in linear disposition. Additional forage (cutting grass) in combination with sugar cane, molasses, or other energizer. | 6 |
| | | Without iSPS or other silvo-pasture systems. Low forage grass diversity and low tree and bush density (<25 individuals ha$^{-1}$). Only one forage grass species. No additional forage. Complemented with mineralized salts. | 3 |
| | | Without iSPS or other silvo-pasture systems. Very low diversity of forage grasses, without trees and bushes. Only one species of grass as monoculture. The trees have been removed from the pastures. Without forage banks. Not complemented by mineralized salts. | 0 |
| Pasture rotation (PaR) | Grazing system, time, measurements | Semi-stabled: The animals spend most of their time confined under a roof. Very short grazing periods (hours per day). | 10 |
| | | Highly rotational in strips or small pastures, isolation with electric fence. Short periods of stay (maximum between 1 to 2 days). Gauges are practiced. Pasture recovers quickly. | 8 |
| | | Moderate amount of rotation in medium-sized pastures, isolated by electric fence or live fence. Medium periods of occupation, between 3 and 7 days. Measurements are not carried out. The pasture is able to recover before the occupation cycle begins. | 6 |
| | | Little rotation in large pastures. Long occupation periods from 8–30 days, isolated or not by live or electric fences. | 3 |

| | | Measurements are not carried out. The pasture is not able to recover until the following occupation period. | |
|---|---|---|---|
| | | Being large in size, no pasture rotation. Occupation periods of more than 30 days. Measurements are not carried out. The pastures do not recover. | 0 |
| Water management (WaM) | Origin, transportation, use, storage, quality control for animal consumption. | Natural sources (sources, glens). Cattle aqueducts for circulation of treated and/or potable water. If there is irrigation, appropriate technologies are used. Frequent (by semester) physiochemical and bacteriological analysis. Total availability and potability. | 10 |
| | | Natural sources that supply the fixed water distribution system (pipes or hoses), with no leakage. If there is irrigation, appropriate technologies are used. Infrequent physiochemical and bacteriological analysis (yearly). Total availability and partial potability. | 8 |
| | | Artificial reservoirs (wells, water harvest, ponds, cisterns) that supply the water distribution system (hoses) with leaks. If there is irrigation, appropriate techniques are used. Infrequent phytochemical and bacteriological analysis (biannual) or none at all. Partial availability and partial potability. | 6 |
| | | Artificial reservoirs. Connection or transport through hoses, with leaks. If there is irrigation, appropriate technologies are used. Infrequent or non-existent phytochemical and bacteriological analysis. Partial availability and apparent partial potability. | 3 |
| | | Artificial reservoirs. Manual transport. No physical–chemical analysis. No guaranteed availability or potability. | 0 |
| Sanitary management (SaM) | Parasite control methods | Control of parasites (ecto and endo) is based on alternative veterinary medicine (food supplements with de-parasitized plants or immune stimulants, baths with repellent plants or minerals, homeopathy, acupuncture). Other complementary practices: preventive coprological exams, biological/natural control of flies and gastrointestinal parasites with dung beetles, co-phages, parasitoid wasps, entomo-pathogenic/ anthelmintic fungus, or others. | 10 |
| | | Parasite control (ecto and endo) is based on the use of alternative veterinary medicine. There are no complementary practices. | 8 |
| | | Parasite control (ecto and endo) is based on the use of chemically synthesized anthelmintic medicines in less than recommended annual doses, and only in animals. | 6 |

| | |
|---|---|
| Chemical substances are used in recommended annual doses in all the herd. | 3 |
| Parasite control (ecto and endo) is only performed with anthelmintic, endectocide, and other synthetic drugs, in yearly doses superior to those recommended. | 0 |

In mainly tropical and subtropical American countries, conventional cattle raising based on the monoculture of foraging species and little presence of tree cover has been one of the important causes for the loss of biodiversity. That is why it is necessary to develop cattle raising practices that favor agro-biodiversity conservation [59]. The need to evaluate use and dosification of veterinary medicines for the control of ecto and endo parasites such as lectonas macrocilicas (mainly, Avermectine and Ivermectine) is of special interest due to the negative effects to the fauna that contribute to recycling bovine manure in the pastures.

Silvo-pasture systems, where arboreal or bushy vegetation is found in different arrangements complementing the consumption of grains for nutrition, have become viable alternatives for many cattle ranchers in the region. Studies by Murgueitio et al. (2009), Vallejo et al. (2010), among others, have documented positive effects of these systems in carbon, nitrogen, organic matter, pH, enzymatic activity, and soil microorganism biomass values as well as positive effects in biological control of parasitic flies in cattle [35,60]. Besides having trees for timber and human consumption, the presence of threatened flora and tropical birds visiting diversified living fences strengthen producer food security and aggregate values, such as agro- and ecotourism, for the productive system.

Over the past years, the concept and practice of regenerative cattle farming (focused on the agricultural sector as well), the object of which is soil conservation, produce an increase in organic matter (see conservation practice criteria) and contribute to climatic regulation through the assimilation of great quantities of $CO_2$, besides offering the consumer more nutritive and healthy food [61].

In other non-tropical scenarios, ecological, organic cattle raising has been proposed as an alternative to intensive cattle raising. Here, practices that consider animal welfare and their positive consequences in obtaining meat and lactose products free of chemical residue, at the same time contributing to reducing soil and water contamination, are proposed [62,63]. For this, alternative medicines (homeopathy, aromatherapy, phytotherapy) are proposed to treat infection and inflammations, obtaining the certification of good ecological livestock practices as an aggregate value [64].

Once the assessment of the indicators is defined, individual summation is carried out to obtain the total value of agro-biodiversity of the cattle management practices used in Equation (17).

$$lMP = \frac{SoP + SiA + PaR + WaM + SaM}{5} \qquad (17)$$

On many occasions, agro-ecosystems present integrated agricultural and livestock uses. To attend to this situation and to apply management practice criteria, weighted qualification (grade) of both agricultural and cattle management practices is proposed (Equation (18)).

$$MP = \frac{a_a aMP + a_l lMP}{a_a + a_l} \qquad (18)$$

where $MP$ = qualification of management practices, $a_a$ and $a_l$ = areas destined as agricultural and cattle lands, respectively, $aMP$ and $lMP$ = qualifications for agricultural management practices and cattle management practices.

On evaluating the indicators, from the practices that favor and do not favor agro-biodiversity and interpreting this assessment, it is possible to identify different types of

management systems. Those with an average from 6.0 to 10 would be related to ecological, sustainable, alternative or regenerative agricultural or livestock systems. Those with an average from 4.0 to 6.0 would be related to agricultural or livestock systems in transition. An average from 0.0 to 4.0 would generally represent conventional agricultural or livestock systems [12].

Management practices and other cultural criteria can be evaluated using qualitative methodologies that collect and compile the concepts of agro-ecosystem owners and administrators. These methods include participant observation, structured and semi-structured interviews, polls, field diary notes that allow a reconstruction of every day practices, intentions, motivations and values underlying producer or owner farm management [65], and an evaluation can be supported by secondary information, according to case. The application of polls or interviews that allow for free expression by the farmer, using a thematic axis defined by the interviewer, can be analyzed using tools such as ATLASti or other discourse analysis software [66].

*3.8. Conservation Practices (CP)*

Conservation of natural elements within the agro-ecosystem is also inherent to traditional forms of agriculture. These systems of traditional knowledge linked to conservation and management of biodiversity have been fundamental for their coexistence throughout time [67,68]. That is why conservation practices are fundamental to an integral description of agro-biodiversity and the evaluation and understanding of how they can aid the productive system (Table 5).

Soil conservation practices that complement those carried out in conservation agriculture and that are not directly related to production are considered here, as those that contribute to containing and physically avoiding erosion or detachment of blocks of earth, such as protection of slopes and construction of terraces, and they are appropriate and easy to carry out methodologies [69].

The absence of these conservation practices can cause or increase edaphic erosion, especially under protective coverage where land use conflicts exist. Where there is extensive cattle farming on hillsides, it is common to find evidence of medium to severe erosion where grooves, bald spots, terracettes and gullies have been caused by the continuous stamping of cattle [70,71].

Water, tied to the soil component, is a vital resource for food production and crop productivity, and depends, among other things, on the quantity of organic matter present. In greater concentrations of humus and other decomposing organic substances, the capacity to retain water is increased [72,73].

In regions where water is naturally scarce or where soils have suffered erosion and the bio-structure has decreased or been lost, soil management requires capital investment and hydraulic infrastructure in order to obtain efficient use. In regions with technified agriculture, the development of new, intelligent irrigation systems is frequent (transferred into commercial uses through patents) to maximize their use [74,75]. When financial resources are scarce, appropriate technologies such as micro-uptake, ridges, terraces, contour plots, water harvesting, domestic aqueducts or reservoirs, among others, are used [76,77].

According to current regulations (in the case of Colombia, Ministry of Agriculture Decree 1449 of 1977 stipulates that water sources must be isolated and surrounded by protective vegetation at least 100 m from its perimeter and river ravines, and there must be approximately 30 m of protective forest on each side of the channel), if water channels cross the farms, or if they have their own sources of water or aquifers, it is best to maintain their protective forests.

**Table 5.** Descriptors, evaluation and values for conservation practices indicators (CP) (Source: modified by [12]).

| Indicator | Description | Evaluation Categories | Value |
|---|---|---|---|
| Soil Conservation Practices (CsP) | Erosion control methods. Soil analysis. Use conflicts | Use of at least three erosion control methods, overgrazing control, slope protection, construction of terraces, carving or gabions. There is no soil use conflict. Carry out periodical soil analysis. | 10 |
| | | Use one or two methods of erosion control. Irrigation using appropriate technology. Carry out, or not, periodical soil analysis. Conflict over soil use exists on at least 25% of the farm area. | 8 |
| | | Use of at least one method of erosion control. No soil analysis. Conflict over soil use on a part of the farm (between 25 and 50% of the area). | 6 |
| | | No use of erosion control methods. No soil analysis. No use of erosion control methods. No soil analysis carried out. Irrigation using Inappropriate technologies. Conflict in soil use on most of the farm (between 50 and 75% of the area). | 3 |
| | | No use of erosion control methods. No soil analysis. Irrigation using Inappropriate technologies. Conflict in soil use in over 75% of the farm area. | 0 |
| Water Conservation Practices (CwP) | Protection of bodies of water. Water collection. Hydric balance. Spills | Sources, recharge sites and rounds of stream, ravines, rivers, and protected bodies of water with natural vegetation according to environmental regulations. Carries out water collection practice: water harvesting, recycling, deviation ditches, jagüeyes, wells, reservoirs, if necessary. Use hydraulic balances. No contaminating spills. | 10 |
| | | All water sources or springs protected by natural vegetation, without following environmental regulations. Some water collection practices when necessary. No contaminating spills. No hydraulic discharges. | 8 |
| | | Water source(s) or spring(s) are 50% or more protected by natural vegetation. Few water collection practices. No contamination of water spills. | 6 |
| | | The hydric source(s) or spring(s) are at least 50% totally protected by natural vegetation or barbed wire. Some complementary practices. Contaminating spills. | 3 |
| | | No protected spring. No complementary practices. Contaminating spills. | 0 |
| Biodiversity Conservation Practices (CbP) | | Six of the following practices are found: reforestation with native species, management of other covers for natural recovery, intentional introduction of native or useful species (plants with flowers and fruit, plants—trap, medicinal and | 10 |

| | |
|---|---|
| aromatic), habitat protection for various animals, germplasm banks. | |
| Evidence of 4 to 5 of the practices mentioned. | 8 |
| Evidence of 2 to 3 of the practices mentioned. | 6 |
| Evidence of at least 1 of the practices mentioned. | 3 |
| No use of biodiversity conservation practices. | 0 |

Once the assessment of conservation practices is obtained, assessment and interpretation are carried out using Equation (19).

$$CP = \frac{CsP + CwP + CbP}{3} \tag{19}$$

*3.9. Perception, Awareness and Knowledge (PAK)*

Perception is related to the sensorial experiences a subject experiences when recognizing the world [78]. These, at the same time, are interpreted and defined in the framework of cultural guidelines (ideologies, life histories, experiences, political contexts, among others) that are adjusted according to the need for survival and social coexistence through symbolic thought, survival and social coexistence [79]. According to Wilson (1984), many apparently irrational perceptions, attitudes and behaviors of present society regarding biodiversity manifest themselves in the context of the human species′ evolutionary and ecological history [80]. Evaluation and use of nature is therefore predetermined by the adaptive relationship between human beings and natural species, ecosystems, or phenomena.

From the perspective of agro-biodiversity, perception is associated with sensorial experiences that producers, administrators and owners interpret as benefits derived from biodiversity such as food and water supply, soil fertility or the generation of microclimates for crops, among others [81].

Consciousness or awareness connects the subject with their interior and exterior world, giving them a sense of their reality and adding to a sensation of transcendence [78]. This, in the field of MAS, means recognizing the importance of biodiversity at its different levels of organization (genes, species and ecosystems), timescales (humans, biological, geological and, even, cosmic) and space (from local to planetary), beyond its use value or legacy (individual benefits, family or patriotic motivations, ancestry, among others). It also permits recognizing the connection and interdependence that exist between humanity and the other beings with which we share the planet and appreciating the value of their existence and their role in the web of life. However, there is a sense of individual (or local) responsibility with respect to the whole, globally: an environmental ethic [82–84].

Finally, the knowledge acquired by the farmer, producer or owner permits the application of techniques beneficial to the sustainable conservation, use and management of agro-biodiversity. Sources of knowledge are varied and can also be associated with academic studies and institutions and traditional knowledge originating from local experimentation and adaptation of agroecological practices by native communities, farmers, or peasants [85–87].

Within this criterion, it is also important to exalt the entire framework of symbolic relationships (identitary, ritualistic, valorative, attitudinal, interpretative) that persons and communities weave around agro-biodiversity, especially as a tribute to food and the curing power of medicinal plants. These relationships are not only associated with their bio-cultural heritage and traditions, traditions learned as popular knowledge through oral tradition, but also with social values (solidarity, spirituality, generosity, happiness, and love) [68,88], expressed in their attitudes and actions in agro-ecosystems.

Categories for the evaluation of the two indicators for PAK criteria are presented in Table 6 and Equation (20).

$$PAK = \frac{PeCo + Kno}{2} \tag{20}$$

**Table 6.** Descriptors, evaluation and values for perception, awareness and knowledge indicators (PAK) (Source: modified from [12]).

| Indicators | Description | Evaluation Categories | Value |
|---|---|---|---|
| Perception and conscience (PeCo) | Perception—conscience: Level of understanding of the importance (I) of agro-biodiversity, conservation and of the benefits (B) that this offers Perception level depends on expressing both importance and benefits (I-B) and that the discourse accompanies or materializes in concrete actions in agro-ecosystem conservation and management | The farm owners or administrators express both the perceived importance and benefits from agro-biodiversity in agro-ecosystems, and this double character materializes in well-defined management and conservation actions. | 10 |
| | | The farm owners and/or administrators express both perceived importance and benefits received from agro-biodiversity in their agro-ecosystems, but they only materialize in actions in one of the two aspects. | 8 |
| | | The farm owners and/or administrators express the importance or benefits of biodiversity, but not an I-B relationship. There are no concrete actions in their agro-ecosystems to support their words. | 6 |
| | | The farm owners and/or administrators express the benefits but not the importance of agro-biodiversity. There are no concrete actions in their agro-ecosystems to support their words. | 3 |
| | | The farm owners and/or administrators do not express either the importance or the benefits they obtain in their agro-ecosystems. They show no interest in the topic. | 0 |
| (Kno) | Knowledge: degree of conceptual clarity regarding components of agro biodiversity and a notion of the underlying processes of structural connectivity of agro bio-diversity in order to potentiate their relations and functions in the productive system, | The owners and/or administrators are familiar with specific components of biodiversity (plants, animals, fungus, and other microorganisms) present on the farm, as well as uses, properties and other popular knowledge. They also know the role of vegetation connector methods to potentiate agro-biodiversity and the productive system. | 10 |
| | | The owners and/or administrators are familiar with certain specific components of bio-diversity (ex. plants, animals) but have very little notion of | 8 |

| | | |
|---|---|---|
| acquired through academic or technical education, or learning from life (popular knowledge) | vegetation connectors or methods to potentiate the benefits of agro-biodiversity. | |
| | The owners and/or administrators are familiar with few components of biodiversity (Ex. plants) and have some knowledge of the role associated with vegetation connectors. They recognize certain methods, but not the benefits to their system. | 6 |
| | The owners and/or administrators are familiar with few specific biodiversity components (Ex: plants) and have some related knowledge, but not of the role of vegetation connectors. They know of no method to potentiate the benefits of agro-biodiversity in their productive system. | 3 |
| | The owners and/or administrators do not recognize any specific component of biodiversity or knowledge associated with the role vegetation connectors or of methods to potentiate biodiversity in their productive system. | 0 |

*3.10. Action Capacity (AC)*

In order to increase and maintain agro-biodiversity on farms, it is vitally important to produce concrete actions that, at the same time, guarantee the wellbeing of the proprietors and the groups that produce and organized them. These actions depend on certain abilities that the owner and the community where they are located have or may have developed.

In order for the efforts and processes that contribute to enriching farms with species and diversifying them with agro-biodiversity management and conservation practices, much financial muscle (saved capital) or, in its defect, debt capacity (access to credit), is required. Areas with or without much productive potential must also be transformed into forest conservation areas or diversified with permanent crops that give long-term results [89,90]. In some townships, counties, or rural areas, there are regulations that encourage forest or water conservation by the extension or reduction of land tax payments. This condition can facilitate cover conservation practices on the properties [91]. Added to the above, the logistics surrounding access to markets and supplies for strengthening cover and agro-biodiversity, in general, as well as vital infrastructure, transport and labor, are conditions that permit the consolidation of agro-ecological production practices and sustainable daily activities [92–94].

In this same sense, management capacity is an exercise that permits mobilizing forces for strengthening agro-biodiversity on the farms. These can come from both the institutional sector or from community initiatives toward a common ecological objective. This also requires conscious planning of soil use from an ecological perspective. An aspect worthy of special attention is the management of and access to markets for organic or agro-ecological products that guarantee fair prices [94,95].

In countries such as Colombia, it is also fundamental to take fractioning or destruction of the social networks into account in areas where there is armed conflict, public security problems and violence that can reduce the organizational and logistic capacity of the small farmer or small producer to commercialize their products, as well as the capacity

of the markets to guarantee a continuous food supply for those who wish to purchase. Exercises in support of local development and the strengthening of these productive sectors (the creation of peasant agro-ecological markets or participative guarantee systems (PGS), for example) are ways of contributing to increasing farmer capacity for action and consumer decision during or after a cease in hostilities [96,97].

Finally, access to appropriate adequate technologies conditioned for agricultural activities, long cycle species, agro-ecological production, conservation and, even, financial, logistic and administrative management, technical accompaniment and training on topics related to establishment and management are basic tools for owners and administrators to strengthen farm agro-biodiversity and productivity [98,99]. Table 7 and Equation (21) present the evaluation categories for the criteria.

$$PAC = \frac{EfC + LoC + MaC + TtC}{4} \tag{21}$$

**Table 7.** Action capacity value descriptors (AC) (Source: modified from [12]).

| Indicator | Description | Categories of Evaluation | Value |
|---|---|---|---|
| Economic and financial capacity (EfC) | Destination of financial resources for coverage conservation, and natural resource and agro-ecological processes A: Savings and personal resources AC: Access to credit AP: Access to institutional support programs (ASP, support from NGOs. Land tax exemption, among others) DA: Destination of areas with productive potential (agricultural or livestock systems) to conservation | A, AC and AP used as sources of financing in processes of coverage improvement and agro-ecological production. There is also the possibility of counting on the possibility of changing a productive use to a conservation use (DA). | 10 |
| | | Two of the three sources of financing directed toward coverage improvement and agro-ecological production exist. There is also the possibility of changing a productive use to a conservation use. (DA). | 8 |
| | | One of the three sources of financing directed toward coverage improvement processes and agro-ecological production are present. There is also the possibility of changing from a productive use to a conservation use (DA). | 6 |
| | | No external sources of financing are present, but there is the possibility of changing from a productive use to a conservation use (DA). | 3 |
| | | There are no external sources of financing or possibility of changing from a productive use to a conservation use (DA). | 0 |
| Logistic capacity (LoC) | Conditions of mobility, availability of qualified labor to work in strengthening of vegetal cover processes and agro- | There are good access roads, good access to means of transportation. There are nearby nurseries and readily available labor for strengthening vegetal cover and/or agricultural production. | 10 |
| | | Three logistic conditions required for strengthening vegetal coverage are present. | 8 |

| | | | |
|---|---|---|---|
| | ecological/sustainable production | Two logistic conditions required for strengthening coverage are present. | 6 |
| | AMT: Access to means of transportation | One logistic condition required for strengthening coverage is present. | 3 |
| | FN: Nearby forest nurseries LA: Labor availability | No logistic condition required for strengthening coverage are present. | 0 |
| Management capacity (MaC) | Farm management factors to improve and strengthen vegetal cover, promote agro-biodiversity and production and agro-ecological /sustainable marketing RI: Relations with institutions Associability or capacity to form alliances with the community PP: Shows planning of soil uses MC: markets for commercialization | The four management faction oriented toward maintaining vegetal cover and agro-ecological production are present. | 10 |
| | | Three management factors oriented toward maintaining vegetal cover and agro-ecological production are present. | 8 |
| | | Two management factors oriented toward maintaining vegetal cover and agro-ecological production are present. | 6 |
| | | One management factor oriented toward maintaining vegetal cover and agro-ecological production are present. | 3 |
| | | No management factor oriented toward maintaining vegetal cover and agro-ecological production is present. | 0 |
| Technological and Technical Capacity (TtC) | ATc: access to adequate/appropriate technology TA: technical assistance in ecological/sustainable agriculture or livestock, and conservation of natural resources CA: offer of training in topics of ecological/sustainable agriculture or livestock and conservation of natural resources | Access to appropriate or adequate technologies for field work. There is frequently an offer of technical assistance and the presence of development institutions oriented toward agro-biodiversity or ago-ecological production. | 10 |
| | | Access to appropriate or adequate technologies. There are infrequent offers of technical assistance. There are development institutions offering programs oriented toward agro-biodiversity or agro-ecological production. | 8 |
| | | There is no access to appropriate or adequate technologies. There are technical assistance offers. There are no programs oriented toward agro-biodiversity or agro-ecological production. | 6 |
| | | There is no access to appropriate or adequate technologies. There are offers of technical assistance although infrequent or not well directed. There are institutions that give infrequent support to agro-bio diversity or agro-ecological production programs. | 3 |

| | |
|---|---|
| There is no access to appropriate or adequate technologies; no offer of technical assistance. There are no institutions that promote programs oriented toward agro-biodiversity or agro-ecological production. | 0 |

After assessing the indicators, they are added to the respective criteria, obtaining a maximum individual assessment of 10 in each of them. At the same time, the criteria are joined, added or weighted (according to the decision of the researcher) according to Equations (22) and (23), respectively, and a final result is obtained varying from 0 to 100. This value defines how developed or how agro-biodiverse the MAS is, and it is interpreted according to Table 8.

$$MAS = CMLS + EEC + EIC + DEC + DIC + LU + aPM, lPM + CP + PAK + CA \qquad (22)$$

The initial additive estimate of the EAP index can also consider differential weights for each criterion according to application needs. Those weighted $\beta_i$, with I = 1 to 10 (a coefficient for each criterion represents a weight given to each of them and can originate from the concensus obtained in the evaluation process, consultation with experts, or studies that have established values more adjusted to specific needs). It is important to remember that in comparative studies, those weighted must be the same for all agroecosystems.

$$MAS = \beta_1 CEEP + \beta_2 ECE + \beta_3 DCE + \beta_4 ECI + \beta_5 DCI + \beta_6 US + \beta_7 PMa, PMg + \beta_8 PC + \beta_9 PCC + \beta_{10} CA \qquad (23)$$

**Table 8.** Interpretive scale for the Main agro-ecological Structure (MAS) of the farm (Source: [12]).

| Numeric Value | Interpretation |
|---|---|
| 91–100 | Completely developed in their agro-biodiversity. |
| 81–90 | Very strongly developed in their agro-biodiversity. |
| 71–80 | Strongly developed in their agro-biodiversity. |
| 61–70 | Moderate to strongly developed in their agro-biodiversity. |
| 51–60 | Moderately developed in their agro-biodiversity. |
| 41–50 | Slightly to moderately developed in their agro-biodiversity. |
| 31–40 | Slightly developed in their agro-biodiversity. |
| 21–30 | Weakly developed in their agro-biodiversity. |
| 11–20 | Very weakly developed in their agro-biodiversity. |
| <10 | With no structure or agro-biodiversity. |

## 4. Discussion

From an environmental point of view, this proposal integrates not only biological and ecological variables, but also cultural. These represent both the agro-biodiversity present in the agro-ecosystem of each farmer or owner and its forms of management and conservation in the situations, histories, values and beliefs of each farmer or owner [12].

The advantages supposed by MAS applicability in different contexts and agro-ecosystems are found in the flexibility of the methodological tools proposed to describe indicators based on simple methodologies such as social cartography and the characterization of the different types of vegetables with farmer support, as well as the possibility of using

geographic information systems and hiring a professional botanist to characterize the floristic and structural composition of the vegetation. In the same manner, social methodologies to describe cultural indicators can start with detailed observation and accompaniment and dialogue with the farmer, later complemented by the use of software for the analysis of qualitative data. Everything will depend on budget, an established human team, and project scope [12].

The calculations for evaluation and interpretation are simple and can easily be carried out by owners, technicians, students, professors or decision makers. This can strengthen a methodology based on a dialogue of knowledge, which will enrich the research and the participating researchers.

Additionally, this operational ease can permit evaluation based on the creation of a systematized operative tool (digital application) that can be downloaded into mobile devices located in any rural large data base and that can also feed a platform for free or institutional use (with uses for territorial planning) once the owner decides to share their information. Factors that can limit the possibility of this tool having generalized use would mainly be the verification of quality of information supplied and 3G and 4G signal coverage.

It is important to emphasize that the objective of the index is that of universal application. However, it is also necessary to considerer some aspects of measurement and adapt indicators and assessment ranges to certain particularities of the zones studied. For example, in the present case, assessment ranges of connector diversity criteria (III and V) were adapted to the wealth and vertical structure typical of the premountain forests of the Andean region of Colombia, a result of its particular physiographic and climatic conditions [27].

In the neo-tropical region, as cattle raising is one of the most representation productive lines [35], ways of managing agro-biodiversity must be considered in this proposal. In other regions, pisciculture and goat, pig and poultry raising can be more important in farm production, making it necessary to produce criteria for evaluating these management practices. This argument can also be applied to certain traditional cultivation practices, such as the so-called "slash and burn" migratory practice that can be criticized from the point of view of soil and forest conservation, but that expresses other realities and possibilities for local farmers. The index could be adapted to collect these particularities.

Other cultural indicators could also include discussion, adaptation or change according to local contexts, given the various ways of individually or collectively thinking or deciding about territories. Some social organizations are based on the active participation of their members; other ethnic groups such as the Nasa in the Department of Cauca, Colombia, base their agro-ecological knowledge on bottom to top organizational relations, assistance that is projected through the home gardening promoters (tul promotor), guided by an indigenous community council (cabildo). Where State and market have little effect, action criteria ought to weigh these community and organizational relationships and types of technical and technological capacities.

It is also important to mention its disadvantages although they do not demerit its applicability and usefulness. As with all indexes, it is a quantitative and qualitative aggregate of indicators to produce a single value capable of producing a simplistic vision of the attribute or system studied and to possess different units of study, according to the indicators, making it difficult to weigh them and having a certain degree of subjectivity due to the transformations or normalizations of the values measured in each indicator [100].

According to Beaver and Belloff (2000), the metrics should be simple, easy to evaluate, understandable, cost-effective, reproducible, robust, not contradictory, and useful for making decisions, among others [101]. Some MAS indicators, especially those contained in management practice criteria (MP), contain several variables within the same indicator, reducing its simplicity and clarity to permit subjectivity. They were designed for the purpose of collecting the greatest number of agro-bio-diverse practices in each productive phase that could increase productive system biodiversity.

To overcome these "limitations", researchers must:

1. Understand each of the indicators (and their variables) and identify their importance according to the objectives of the specific study. Understanding the cause would permit the elimination of certain related variables that overcomplicate the indicator but do not compromise the intention.

2. Add, in a weighted manner, the indicators built with the same unit of measurement, within the criteria, as was proposed for the aggregation of criteria in the index (see Equation (23)). It is also desirable to combine complementary methodologies to "emphasize" the importance of the indicators that may "hide" behind the final evaluation of the index. Quantitative methods such as the AMOEBA diagrams [102] and qualitative methods such as Design Structure Matrix (DSM) [103] permit visualizing the state of the different indicators in the evaluation scale constructed and in those that structure the system called MAS. Multivariate tests can be another alternative for interpreting the importance of certain indicators compared to others such as principal component analysis (PCA) [104] that collect variability in a few dimensions or main components, reflecting which indicators most contribute to this conformation and to selecting the model according to its adjustment.

Some MAS studies in the context of agroecology and environmental analysis in Colombia are kept in the repository of the National University of Colombia´s main library (https://repositorio.unal.edu.co/). This research evaluates index behavior in coffee and citrus agroecosystems and its relation to resilience to climate change (https://repositorio.unal.edu.co/handle/unal/58147; https://repositorio.unal.edu.co/handle/unal/63924, (accessed on 12 June 2022).), compared to the diversity of Andean tubers cultivated on traditional small-farm agroecosystems and associated ancestral knowledge (https://repositorio.unal.edu.co/handle/unal/75774; https://repositorio.unal.edu.co/handle/unal/76482, Accessed on 12 June 2022). This permits recognizing and quantifying changes in the structures of territories and rural landscapes (https://repositorio.unal.edu.co/handle/unal/52347 [87] (accessed on 12 June 2022), the visit of bees to coffee plantations [105], as well as those mentioned in Leon-Sicard (2021) [12]. Since the original formulation of the index by Leon-Sicard [15], these studies have led to the improvement of criteria and indicators culminating in the final version presented here. The authors firmly believe that, as divulgation of the index increases, researchers can revise and adapt the index to their specific conditions.

Furthermore, the description of agro-system MAS acquires greater importance when connecting the farm with the landscape through agro-biodiversity. In these so-called rural landscapes, the matrix is formed by a particular type of anthropic cover or mosaic of productive systems with their own cultural and biological characteristics [91]. However, when increasing the appreciation scale, a rural landscape is, at the same time, composed of larger units: farms. According to Hart (1985) [106], the farm is a unit of agricultural production, and according to [14,15], it is the space where complex relations meet among ecosystems, forms and culture: in other words, the agro-ecosystems.

In this sense, the farm is a minimal spatial unit in a rural territory where productive processes take place and where all the properties, ecological and cultural elements of a landscape interact. In this vision, farms where the different habitats or elements of the landscape are found could be considered as units of agro-biodiversity analysis on which the evaluation of more holistic approaches could be built, using both approaches from landscape ecology and metrics that consider factors of a cultural order, in order to understand how it is structured.

This need for interdisciplinary approaches in the evaluation of agro-biodiversity was recently made manifest by the FAO (2018, 2019), the entity that has concluded that, to carry out this purpose, the qualitative methodologies of the social disciplines such as anthropology, ethno-biology, and biogeography (interviews, focal groups, and participative methodologies) must accompany the already traditional quantitative methodologies used

to evaluate biological diversity belonging to the fields of genetics, ecology and conservation biology [25,107–109].

The MAS becomes an environmental index that groups both ecosystemic and cultural dimensions and some of its methodologies in order to understand how agro-biodiversity is structured, principally locally, but also recognizing that landscape has a modeling effect. At the same time, the farms as a group and the decisions their owners make mold the agro-landscape. This is why the MAS also permits materializing and making visible these smaller scales that are used in territorial ordering, both for the farms and for the farmers whose decisions are fundamental in organizing the territory [12].

Here, in order to understand how to structure it, approaches that consider cultural factors from both metrics and landscape ecology would be used.

## 5. Conclusions

The Main Agroecological Structure of agroecosystems is an environmental index since it includes ecosystemic and cultural criteria that reveal some of the principal relationships established among human groups (farmers) and their biophysical surroundings.

The index centers on the quantitative and qualitative measurement of agrobiodiversity, especially in terms of structure. Their determination can serve as the basis for studies that include the functionality of this agrobiodiversity.

The index does not replace other approximations and analysis for the understanding of agroecosystem complexity. It is a multidimensional evaluation of agrobiodiversity that, although limited, could complement detailed approaches to the sustainability of agroecosystems and become a part of other farm evaluation or qualification exercises, either individually or in their spatial aggregates of matrices.

MAS is clearly inserted in the intersection between agroecological science and landscape ecology, visualizing one of its fundamental components: farms and their rural proprietors (peasants, farmers, agricultural enterprises).

Based on MAS, studies can be carried out in conservation biology, territorial ordering, ecosystemic services and diversity management, among many other applications.

**Author Contributions:** All authors have contributed to the different stages in the elaboration of the document. All authors have read and agreed to the published version of the manuscript.

**Funding:** This research was funded by the German Academic Exchange Service (DAAD) through the Doctoral Studies Support Program between the Institute for Development Research of the University of Bonn (ZEF) and the Environmental Studies Institute of the National University of Colombia (Instituto de Estudios Ambientales de la Universidad Nacional de Colombia-IDEA/UN), and the Colombian Ministry of Science and Technology (Colciencias).

**Acknowledgments:** Thanks to Álvaro Acevedo, Juliana Sabogal, Mercedes Murgueitio and Elisabeth Jimenez, for their comments and contributions to the redefinition of certain indicators considered in the MAS.

**Conflicts of Interest:** The authors declare no conflict of interest. The funders had no role in the design of the study; in the collection, analyses, or interpretation of data; in the writing of the manuscript; or in the decision to publish the results.

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
