# Peer review of "Main Agro-Ecological Structure: An Index for Evaluating Agro-Biodiversity in Agro-Ecosystems"

_sustainability, doi:10.3390/su142113738_

Round 1

Reviewer 1 Report

Dear authors and editor,

This is a good research that evaluates the Main agro-ecological structure, MAS: an index for evaluating agro-biodiversity in agro-ecosystems.

Research was well designed and detailed results were presented. However, some aspects are not very clear, it should be further improved for possible publication.

Specific comments are:

1. The abstract presented mainly present background, method, and implication, but the results and main conclusions were neglected.

2. The content of tables no. 1 and no.2 is not readable, I recommend their editing.

3. Conceptual and methodological framework includes the results obtained. Even if editor gives the possibility that the paragraph is unique, I would prefer that the be separated from the conceptual and methodological for a clear and easier understanding for the reader.

4. I recommend the authors to include the conclusions section.

Author Response

Dear reviewer.

We would like to thank you for your review and suggestions that will certainly improve the first version of the manuscript submitted to the journal. Please find our answers to each of them either in the text or directly in this letter.

We will be attentive to your feedback or to new suggestions if you consider so

Best regards,

Ingrid

  1. The abstract presented mainly present background, method, and implication, but the results and main conclusions were neglected.

The summary was modified according to your request.

      2.  The content of tables no. 1 and no.2 is not readable, I recommend their          editing

The size of the tables has been increased.

    3. Conceptual and methodological framework includes the results obtained.   Even if editor gives the possibility that the paragraph is unique, I would     prefer that the be separated from the conceptual and methodological for a     clear and easier understanding for the reader.

For better understanding the title of Conceptual and Methodological framework was changed from to Indicators selection and a description of the selection process was included.  In addition, the development and results of the index  are explained in a separate section.

    4. I recommend the authors to include the conclusions section.

Conclusions were included.

Reviewer 2 Report

Dear Autors

The authors of the study "Main agro-ecological structure, MAS: an index for evaluating agrobiodiversity in agroecosystems" attempted to develop 29 indicators of ecosystem biodiversity. The paper also presents the possibility of adjusting the indicators to ecological contexts. Thanks to the great contribution of the authors, the entire material has been carefully prepared. After getting acquainted with the material, the reader may decide that the review of the problem literature has been prepared very well and is extensive. Nevertheless, I have some comments that require clarification.

The reader may find it difficult to determine on the basis of which data the indicators were developed. In the text, I found modest information on / t only in two workplaces. As expected, the material for the analysis comes from Colombia. However, we still have too little information on the subject - it should be completed in detail. Additionally, it is necessary to answer the question: to what extent is this material reliable, is the amount of data to determine the indicators surely sufficient. A question arises regarding the validation of this material with other results from a more distant region. Can the developed indicators be adapted to the region with different characteristics? Or are they not so universal? This information was missing in the text and the summary. The summary of the work or conclusions was not posted at all.

Author Response

Dear reviewer.

We would like to thank you for your review and suggestions that will certainly improve the first version of the manuscript submitted to the journal. Please find our answers to each of them either in the text or directly in this letter.

We will be attentive to your feedback or to new suggestions if you consider so

Best regards,

Ingrid

Specific comments

  1. The abstract presented mainly present background, method, and implication, but the results and main conclusions were neglected.

Conclusions were included.

          2. The reader may find it difficult to determine on the basis of which data      the indicators were developed. In the text, I found modest information on / t      only in two workplaces. As expected, the material for the analysis comes       from Colombia. However, we still have too little information on the subject - it   should be completed in detail. Additionally, it is necessary to answer the question: to what extent is this material reliable, is the amount of data to determine the indicators surely sufficient. A question arises regarding the validation of this material with other results from a more distant region. Can the developed indicators be adapted to the region with different characteristics? Or are they not so universal? 

         The summary of the work or conclusions was not posted at all.

Although the information was clarified in the text, we would like to make the following observations. The MAS originated from the reflexion on the environmental components (ecosystemic and cultural) that a good agroecosystem descriptor should have (Leon-Sicard 2014; 2021 [15; 12] and Leon-Sicard et al., 2018 [13]).  Without publications, but in the form of theses, located in the repository of Universidad Nacional de Colombia´s library, several students and researchers have made theoretical and practical contributions to improve the index. A compendium of these contributions and some of the main contributions are published in Spanish in [12].

In Colombia is difficult obtain funding to do research and to publish, This has limited the possibility the index had previous reception outside the country. However, at the national level, the index (named Estructura Agroecológica Principal) has been recognized by students and some local authorities. The authors hope that, with this and other similar publications, it will increase its acceptance in other countries. For now, an application of the use of drones and MAS [37], was published a couple of days ago and is being well received.

We are aware that, as its dissemination expands, other colleagues will be able to review and adapt the index to their particular conditions. The cases of temperate countries with a high degree of technological development in their agriculture would be very interesting for us.

The summary was improved.

Reviewer 3 Report

I suggest delete MAS in the title.
Line 14: Choose another word instead of proposal, some word like study, research, work, manual, indicators, and Instructions.
Lines 14-18: A too long sentence and difficult to understand, divide it to 2 sentences.
Line 18: insert MAS in parentheses.
In the abstract, there is no mention of the results of the presented indicators and their evaluation. Furthermore, I could not find any content about the MAS test and how much it can be adapted to any environment and conform to different climatic conditions.
Line 22: At the end of the abstract, a summary of the activities taken in this article and also the suggestions of the authors should be presented.
Line 29: No need to define agro-biodiversity in footnote. You presented a long definition, just a short mention in the main text is enough.
Lines 54-55: Previous comment also about large agro-ecosystems.
A conclusion or research gap or future perspective for concluding your activities and suggestion is needed at the end of manuscript.

Author Response

Dear reviewer.

We would like to thank you for your review and suggestions that will certainly improve the first version of the manuscript submitted to the journal. Please find our answers to each of them either in the text.

We will be attentive to your feedback or to new suggestions if you consider so

Best regards,

Ingrid

Round 2

Reviewer 2 Report

Dear Authors,

Thank you very much for taking into account the comments and suggestions and for providing explanations.

Best regards